# Model Uncertainty Analysis Methods for Semi-Arid Watersheds with Different Characteristics: A Comparative SWAT Case Study

**Lufang Zhang [1,2], Baolin Xue [1,2,*] , Yuhui Yan [1], Guoqiang Wang [1,2,*] , Wenchao Sun [1,2], Zhanjie Li [1,2], Jingshan Yu [1,2], Gang Xie [3] and Huijian Shi [3]**

[1] College of Water Sciences, Beijing Normal University, Beijing 100875, China; 201721470030@mail.bnu.edu.cn (L.Z.); 201821470032@mail.bnu.edu.cn (Y.Y.); sunny@bnu.edu.cn (W.S.); lzhan@bnu.edu.cn (Z.L.); jingshan@bnu.edu.cn (J.Y.)
[2] Beijing Key Laboratory of Urban Hydrological Cycle and Sponge City Technology, Beijing 100875, China
[3] Shandong Academy of Environmental Planning, Shandong 250101, China; xiegang888@163.com (G.X.); 13791090303@163.com (H.S.)
* Correspondence: xuebl@bnu.edu.cn (B.X.); wanggq@bnu.edu.cn (G.W.); Tel.: +86-10-58802736 (B.X.)

**Abstract:** Distributed hydrological models play a vital role in water resources management. With the rapid development of distributed hydrological models, research into model uncertainty has become a very important field. When studying traditional hydrological model uncertainty, it is very common to use multisite observation data to evaluate the performance of the model in the same watershed, but there are few studies on uncertainty in watersheds with different characteristics. This study is based on the Soil and Water Assessment Tool (SWAT) model, and uses two common methods: Sequential Uncertainty Fitting Version 2 (SUFI-2) and Generalized Likelihood Uncertainty Estimation (GLUE) for uncertainty analysis. We compared these methods in terms of parameter uncertainty, model prediction uncertainty, and simulation effects. The Xiaoqing River basin and the Xinxue River basin, which have different characteristics, including watershed geography and scale, were used for the study areas. The results show that the GLUE method had better applicability in the Xiaoqing River basin, and that the SUFI-2 method provided more reasonable and accurate analysis results in the Xinxue River basin; thus, the applicability was higher. The uncertainty analysis method is affected to some extent by the characteristics of the watershed.

**Keywords:** hydrological modeling; method comparison; SUFI-2; GLUE; calibration

## 1. Introduction

A hydrological model is an effective tool for exploring complex hydrological processes and solving practical hydrological problems and is based on a combination of computer technology and system theory [1]. As hydrological model research and application have developed, increasing attention has been paid to the uncertainty of these models. The main sources of uncertainty are the structure of the model itself [2], the input data of the model [3], and the uncertainty of model parameters [4]. The uncertainty caused by the model's own structure needs to be solved inventively. The deviation and absence of model input data are mainly limited by external factors [5]. Research into model uncertainty at present mainly focuses on the uncertainty of model parameters.

The SWAT model is a distributed hydrological model that combines remote sensing (RS) and geographical information system (GIS) technology [6], and takes into account underlying surface conditions and meteorological factors on the basis of precipitation data [7]. It is widely used in the simulation of water quantity, sediment, and non-point-source pollution [8–10]. It has many input

parameters, and its model output is usually determined by multiple parameters. The uncertainty analysis of model parameters is critical, therefore, to research into hydrological model uncertainty [11,12].

Traditionally, preferred watershed hydrological model parameters are typically manual (e.g., trial and error) or are automatically optimized by computer. The manual optimization algorithm has a certain subjectivity, which is related to the experience and training of the hydrologist and the understanding of the structure of the model, and it is time consuming and laborious. To achieve efficient exploration and analysis of model uncertainty, many uncertainty analysis methods have been developed and validated worldwide over the past three decades, including Generalized Likelihood Uncertainty Estimation (GLUE) [13,14], Monte Carlo (MC) [15], Markov Chain Monte Carlo (MCMC) [16], Sequential Uncertainty Fitting Version 2 (SUFI-2) [17], and parameter solution [18]. Of these, SUFI-2 and GLUE are the two most widely used methods for analyzing the uncertainty of hydrological models. SUFI-2 is a comprehensive optimization and gradient search method. It not only can simultaneously determine multiple parameters, but also has the function of overall search. It also considers the uncertainty of input data, model structure, parameters, and measured data, and is suitable for models with complex structures [19,20]. The GLUE method combines the random sampling method of MC and the Bayesian framework. It can also comprehensively analyze model uncertainty caused by model structure, parameter values, data errors, and other issues. The method is simple in principle, easy to operate, and suitable for global analysis. It is a widely used method for studying uncertainty in water environment models [21,22]. This study, therefore, uses the SUFI-2 and GLUE methods to study the uncertainty of the SWAT model.

In recent years, research into the uncertainty of hydrological models represented by the SWAT model has produced many results [23,24]. At the beginning of the study, the researchers used a single uncertainty analysis method to conduct sensitivity analysis and uncertainty evaluation of hydrological models [25,26]. Although the uncertainty method can analyze the uncertainty of the model, it cannot guarantee evaluation efficiency. Comparison of the applicability of different types of uncertainty analysis methods in the same watershed has, therefore, become popular [27,28]. Muleta [29] used the uncertainty method GLUE to analyze the uncertainty of SWAT model parameters in the Big Creek basin, Illinois, USA. The study showed that sediment load had a greater impact on the uncertainty of the SWAT model than runoff uncertainty. Abbaspour [30] compared the five common uncertainty methods (SUFI-2, GLUE, Parameter Solutions method, MCMC, and Importance Sampling) on the Thur watershed, an area 1700 km$^2$ situated in northeastern Switzerland. Abbaspour found that the GLUE method based on a Bayesian algorithm was most suitable for model uncertainty analysis. Uniyal [31] analyzed the uncertainty of river runoff simulation in eastern India by means of the SWAT model. The SUFI-2 and GLUE methods are characterized by their own features and practicability. SUFI-2 is a semiautomatic optimization technique, and it can greatly improve the operational efficiency of complex, structured, and highly computational models. In contrast, GLUE is a fully automated and robust optimization technique, and it is easy to implement. Therefore, for a model with complex structure, SUFI-2 analysis is better, and for a simpler model, GLUE analysis is more efficient. Shivhare [32] and Zhao [33] valued the SUFI-2, GLUE, and ParaSol methods in the Ganga River basin, India, and the Jingchuan basin, China. The results proved that SUFI-2 had a better score in the uncertainty analysis of the basin than the two other methods and was more suitable for complex structural models such as the SWAT model.

In studying the existing model uncertainty, most research has mainly focused on different kinds of uncertainty analysis methods in the same basin. Little work has concentrated on different basins using different methods of uncertainty analysis, ignoring the influence of geographical location, valley basin scale, and different characteristics, which may confer advantages and disadvantages to the results of uncertainty analysis. To enrich the results of this part of the research content, this paper uses two different watersheds as its study areas: the mesoscale Xiaoqing River basin in the plain area and the small-scale Xinxue River basin in the hilly area. First, ArcSWAT2012 was applied to simulate runoff from the two river basins. Parameter sensitivity was then compared and analyzed with a sensitivity

analysis module in the SWAT model. SWAT Calibration and Uncertainty Program(SWAT-CUP) software (Swiss Federal Institute for Aquatic Science and Technology, Dübendorf ,Kastanienbaum, Canton of Zurich, Switzerland.) was then applied, using the SUFI-2 and GLUE methods, on runoff simulation calibration for the uncertainty analysis, to explore the applicability of the SWAT model under different basin characteristics and the uncertainty in model simulation, and to obtain two contrasting methods of uncertainty analysis on the different characteristics of the basin.

## 2. Materials

### 2.1. Overview of Research Area

The Xiaoqing River is located in the south of Shandong province, China, and its geographic coordinates are 116°50′–118°45′ E, 36°15′–37°20′ N. The main cities in the Xiaoqing River basin are Jinan, Zibo, Weifang, Dongying, and Binzhou, and there are 5 other prefecture-level cities, 18 counties, and 182 towns in the basin. The basin topography is dominated by plains. The Xiaoqing River basin covers an area of 10,572 km$^2$, has an annual runoff of 1.27 billion m$^3$, and the river length is 237 km$^2$ [34]. The Xiaoqing River has a semi-arid continental climate [35], with an annual average temperature of about 12–14 °C, annual average rainfall of 646.7 mm, and annual evaporation of 1020.8–2417.4 mm. The vegetation types are mainly deciduous broad-leaved forest, mixed with warm coniferous forest. The land use of the Xiaoqing River basin occupies about 63% of the entire basin, and urban land use accounts for 18% of the total.

The Xinxue River basin is located in Shandong province in the southwest of the Nansi Lake water system. Its geographical coordinates are 116°34′–117°2′ E, 34°27′–35°20′ N [36]. The topography of the Xinxue River basin is mainly mountainous or hilly, and so differs from the Xiaoqing River basin. The Xinxue River basin has an area of 686 km$^2$ and the river length is 89.6 km. It has a semi-arid continental climate, annual average temperature of about 13–14 °C, average annual rainfall of 821.5 mm, and annual evaporation of 975.0–1439.9 mm. Land use in the Xinxue River basin is mainly farmland and grassland, each accounting for 37% of the total; urban land use in the basin is less than 8%. The two basins are shown in Figure 1.

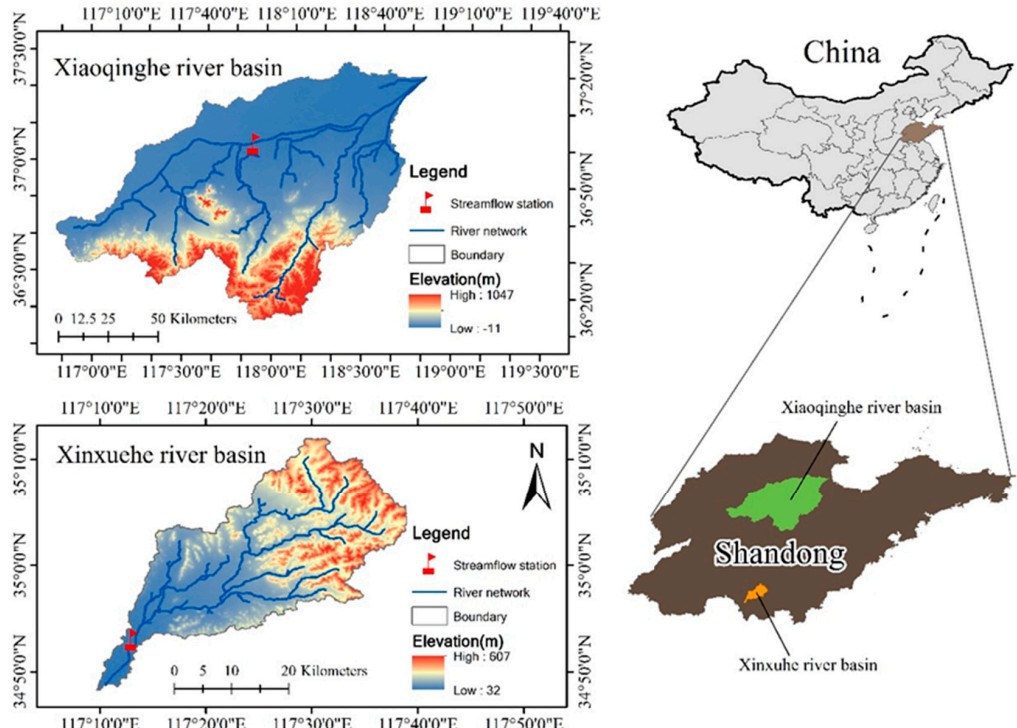

**Figure 1.** Xiaoqing River basin and Xinxue River basin, Shandong province, China.

## 2.2. Data Source

In this study, 30 m resolution digital elevation model (DEM) data and 1:100,000 national land use data from 2010 were used in SWAT modeling, obtained from the Data Center of Resource and Environmental Sciences, Chinese Academy of Sciences. The 1:100,000 soil type map and soil attribute data were from the HWSD Chinese soil data set. There are five meteorological stations (Jinan station, Huimin station, Kenli station, Weifang station, Yiyuan station) in the Xiaoqing River basin and two meteorological stations (Xiji station, Shanting station) in the Xinxue River basin. The measured data for daily precipitation, maximum and minimum temperature, relative humidity, solar radiation, average wind speed, and so forth, from 1985 to 2015 were from the China Meteorological Data Network (http://data.cma.cn/data/index.html). Monthly runoff data from 2006 to 2014, measured at the Chahe hydrological station in the middle reaches of the Xiaoqing River basin and at the Chaihudian hydrological station in the lower reaches of Xinxue River, came from Shandong Academy for Environmental Planning. The model simulation took 2004 and 2005 as the preheating years of the model, using existing basin data, and 2006–2011 as measured runoff data for the preheating years. Using existing basin data, the model simulation took 2004 and 2005 as the warm-up years of the model, 2006–2011 as the rate period of the model, and 2012–2014 as the model validation period.

## 3. Method

### 3.1. Overview of SWAT Model

The SWAT model was developed by the United States Department of Agriculture—Agricultural Research Center. It is mainly based on the Simulator for Water Resources in Rural Basins (SWRRB) model [37] and draws on the main features of Groundwater Loading Effects on Agricultural Management Systems (GLEAMS), Erosion–Productivity Impact Calculator (EP-IC), and Routing Output To Outlet (ROTO) models [38–41]. The SWAT model is used to assist in water resource management by predicting and assessing the impacts of water, sediment, and agrochemical management in untested watersheds. It consists of seven components: hydrology, meteorology, sediment, temperature on the upper soil, crop growth, nutrients, and agrochemicals. From the perspective of model structure, the SWAT model belongs to the second type of distributed hydrological model, that is, applying a traditional conceptual model to each grid unit (or subbasin) to estimate the net rain, then performing the convergence calculation, and finally obtaining the exit section flow.

The SWAT model has been revised many times and produced many versions since it was officially launched in the 1990s. In this study, the ArcGIS (10.2, Environmental Systems Research Institue, Inc., RedLands, CA, USA) interface of SWAT (version 2012, The Agricultural Research Service of United States Department of Agriculture, Washington, DC, USA) has been used, which is popularly known as SWAT. The water cycle simulated by SWAT is based on water balance, which is mathematically expressed as follows [42]:

$$SW_t = SW_0 + \sum_{i=1}^{t} \left( R_{day} - Q_{surf} - E_a - W_{seep} - Q_{gw} \right) \tag{1}$$

where $SW_t$ is the soil water content (mm) at time t, $SW_0$ the initial soil water content (mm), $t$ the simulation period (days), $R_{day}$ the amount of precipitation on the $i$th day (mm), $Q_{surf}$ the amount of surface runoff on the $i$th day (mm), $E_a$ the amount of evapotranspiration on the $i$th day (mm), $W_{seep}$ the amount of water entering the vadose zone from the soil profile on the $i$th day (mm), and $Q_{gw}$ the amount of baseflow on the $i$th day (mm).

The SWAT model has a strong physical foundation and is suitable for large and complex watersheds with different soil types, different agrarian administration modes, and management conditions, and can also be modeled in areas where there is a lack of data. As a result, it has been widely used all over the world [43,44].

### 3.2. Principles and Procedures of the SUFI-2 Method

SUFI-2 is an uncertainty analysis method invented by the Swiss Federal Research Institute of Water Science and Technology [17] for model development. According to this method, parameter values of the model are chosen with the Latin Hypercube (LH) random sampling method within the preset parameter range. The selected parameters are then substituted back into the model, and the value of the objective function is calculated according to the selected objective function. At the same time, the confidence interval 95PPU (95% prediction uncertainty) is calculated by the cumulative distribution of simulation results of 2.5% and 97.5% [45]. The parameter range is the optimal one of the model until the simulation results reach the expected standard; the minimum confidence interval 95PPU includes the most measured data through multiple iterations. This method first determines the objective function, and then gives the initial value range of the given parameters. It then uses LH sampling to obtain n groups of parameter combinations and calculates the objective function of each simulation, then evaluates the simulation effect of each round of sampling with a series of methods, and finally carries out uncertainty analysis [46].

### 3.3. Principles and Procedures of the GLUE Method

The GLUE method was proposed by Beven to evaluate the phenomenon of "isopararism" in model parameters [13,14] and is widely used in the calibration and uncertainty analysis of hydrological models and in other fields. It is important to be aware that the GLUE method, as an analysis of uncertainty, is not a certain parameter in the model but a set of parameters that influences the results of model simulation. Within the preset initial parameter value interval, the Monte Carlo method [47] is adopted to randomly sample the parameter value of the model, and the extracted parameter value combination is substituted back into the model and operated. The likelihood objective function is selected, the value of the likelihood function between the simulated value and the measured value of the model is calculated, and then the weight of these function values is calculated to obtain the likelihood value of each parameter combination. Among all the likelihood values, a critical value is set. The parameter group below the critical value cannot represent the functional characteristics of the model, and the likelihood value of these parameter groups is assigned to zero [48]. The likelihood value of the parameter set higher than the critical value is renormalized. According to the size of the processed likelihood value, the uncertainty range of the model prediction under a certain confidence degree is obtained [47,49].

### 3.4. Evaluation Criteria for Calibration and Uncertainty Analysis Results

For parameter calibration and verification, this study takes the certainty coefficient of determination ($R^2$) and Nash–Sutcliffe efficiency coefficient ($E_{NS}$) for the objective function of the fitting effect and accuracy between the simulated value and the observed value of the two-period model [50,51]. The theoretical value range of $E_{NS}$ and $R^2$ is 0–1, and $E_{NS} < 0$, indicating that the reliability of the simulated value is lower than the real value. The closer $E_{NS}$ and $R^2$ are to 1, indicating the higher the coincidence between the simulated value and the observed value, the better the simulation fitting result [52,53]. Generally, when $E_{NS}$ is greater than or equal to 0.5 and $R^2$ is greater than or equal to 0.6, it is considered that the model can accurately describe the hydrological process in the basin being studied.

With respect to uncertainty of simulation results, this study takes P-factor and R-factor as two measurement criteria. For P-factor, the theoretical value is between 0 and 1, indicating the percentage of simulated data in the observed data within the confidence interval of 95PPU (95% prediction uncertainty, calculated by removing parts less than 2.5% and greater than 97.5% of the cumulative distribution of simulation results). For R-factor, the theoretical value is between 0 and infinity, indicating the average thickness of the 95PPU strip divided by the standard deviation of the observed data. Theoretically, when the P-factor is equal to 1 and the R-factor is equal to 0, the simulated value of the model is exactly

the same as the observed value, while the actual situation is usually that the larger the P-factor value, the larger the corresponding R-factor value, so the value of the two needs to be balanced [54,55].

## 4. Results Analysis

### 4.1. Uncertainty Analysis of Xiaoqing River Basin

#### 4.1.1. Parameter Selection and Range Determination

The value range of sensitivity parameters has a great impact on the results of runoff simulation [56]. The smaller the parameter range, the narrower the uncertainty interval that can be obtained, which improves the confidence level. At the same time, however, it reduces the sensitivity of parameters, making more observation values fall outside the uncertainty interval. In contrast, a larger parameter range widens the uncertainty interval, which can reflect the influence of parameters on simulation results in more detail, but will lead to a decrease in confidence level. Parameter sensitivity analysis and the actual situation in the Xiaoqing River basin allowed us to obtain the sensitivity parameters and value range (Table 1). SUFI-2 was iterated four times and 500 simulations were performed for each iteration. In GLUE, the objective function ENS was set as 0.6, and was iterated twice; 3000 simulations were performed for each iteration. The relevant uncertainty analysis was carried out on this basis.

**Table 1.** Sensitivity parameters and value range of the Xiaoqing River basin.

| Parameter | Parameter Meaning | Initial Value Interval | Affected Object and Process |
|---|---|---|---|
| CN2 | SCS runoff curve coefficient | 35–98 | Surface runoff |
| ALPHA_BF | Base flow alpha coefficient | 0–1 | groundwater |
| ESCO | Soil evaporation compensation coefficient | 0.01–1 | Soil evaporation |
| RCHRG_DP | Permeability coefficient of deep aquifer | 0–1 | Groundwater process |
| GWQMN | Runoff coefficient of deep groundwater | 0–5000 | Soil moisture |
| SOL_AWC | Available soil water | 0–1 | Soil moisture |

#### 4.1.2. Parameter Uncertainty Analysis of the Two Methods in the Xiaoqing River Basin

SUFI-2 was applied to parametric uncertainty analysis in the Xiaoqing River basin (Figure 2). By comparing the degree of each parameter uncertainty and the distribution of the values, it was found that there was an obvious distribution law of ENS value at the scatterpoint that changed with the parameter value, in which the ENS value gradually increased with the overall increase in CN2. ENS is an indicator used to evaluate the results of hydrological model simulations. The greater the ENS value is, the greater the match between the measured value and the analog value is. Therefore, as ENS increased, the uncertainty of parameter CN2 decreased. As ALPHA_BF increased, there was a trend of first increasing and then decreasing; when the parameter value was around 0.5, the ENS value was relatively stable, and the parameter uncertainty was the smallest. As RCHRG_DP increased, the uncertainty increased from uniform distribution to low dispersion, part of the ENS value dropped, and uncertainty increased. The values of the other three parameters were evenly distributed without obvious changes, meaning that these three parameters were more stable and less uncertain. These results show that CN2, ALPHA_BF, and RCHRG_DP had greater uncertainty, and their value change resulted in a great impact on the simulation effect of the model. When the value of CN2 was close to 60, the value of ALPHA_BF was around 0.5, and the value of RCHRG_DP was between 0.07 and 0.08, allowing model simulations to be good.

The GLUE method was applied to uncertain analysis of the SWAT model in the Xiaoqing River basin. The threshold value of target $E_{NS}$ was set at 0.6. Monte Carlo was used to randomly select 3000 groups of samples, and a scatter diagram of parameter values and objective function distribution was obtained (Figure 3). Figure 3 shows that $E_{NS}$ value increased as the values of the three parameters CN2, GWQMN, and SOL_AWC increased. The value of $E_{NS}$ increased and tended to be stable which

means that the uncertainty of these parameters was reduced. As the ALPHA_BF parameter increased, the overall $E_{NS}$ was reduced overall and the degree of dispersion was increased, which led to an increase in parameter uncertainty. The $E_{NS}$ value fluctuated significantly with the change of these parameters, meaning that these four parameters had great influence on the uncertainty of the simulation results. The parameters ESCO and RCHRG_DP were evenly distributed, indicating that the uncertainty was small.

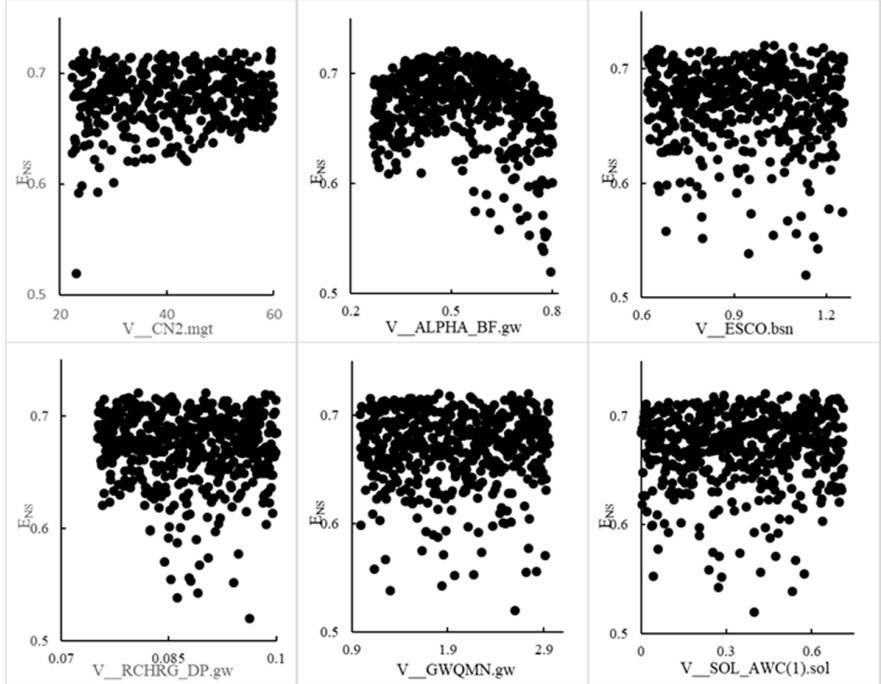

**Figure 2.** Parameter combination and Nash–Sutcliffe efficiency coefficient distribution diagram of the Xiaoqing River basin based on the SUFI-2 method.

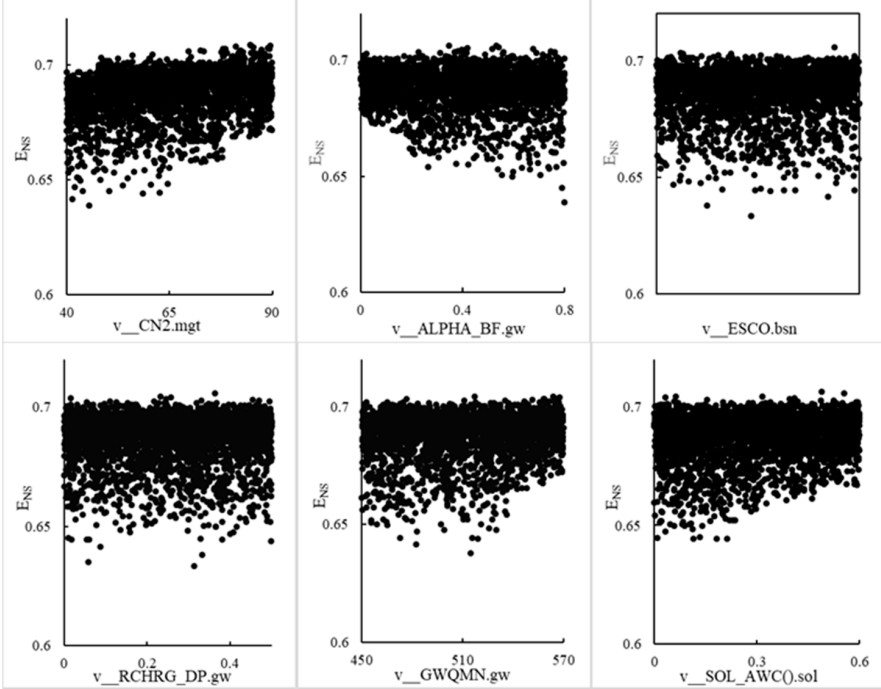

**Figure 3.** Parameter combination and Nash–Sutcliffe efficiency coefficient distribution diagram of the Xiaoqing River basin based on the GLUE method.

### 4.1.3. Analysis of Uncertainty of Model Simulation

After calibration by SUFI-2, the P-factor and R-factor of evaluation coefficient of runoff were 0.77 and 0.75, respectively, during calibration periods, and 0.73 and 0.73, respectively, during validation periods (Figure 4, Table 2). The P-factor in calibration periods was more than that in validation periods, demonstrating that the measured value in calibration periods had more standard deviation than that in validation periods. On average, P-factor was less than 1 and R-factor was close to 0. In calibration and validation periods, $E_{NS}$ was 0.71 and 0.74, respectively, while $R^2$ was 0.72 and 0.76, respectively. These values indicated that SUFI-2 permitted a satisfactory runoff simulation and analysis of uncertainty of the Xiaoqing River basin.

**Table 2.** Comparison of simulation results from the two methods.

| Method | Simulation | P-factor | R-factor | $E_{NS}$ | $R^2$ |
|---|---|---|---|---|---|
| SUFI-2 | calibration period | 0.77 | 0.75 | 0.71 | 0.72 |
| | validation period | 0.73 | 0.73 | 0.74 | 0.76 |
| GLUE | calibration period | 0.84 | 0.80 | 0.71 | 0.75 |
| | validation period | 0.81 | 0.75 | 0.71 | 0.76 |

In this research, we chose 0.6 as the critical value to decide whether the parameter was "useful" for the GLUE method. The parameter and likelihood values were considered to be 0 or normalized by comparing ENS and 0.6. During calibration and validation periods, the P-factor was 0.84 and 0.81, respectively, and the R-factor was 0.80 and 0.75, respectively (Figure 5, Table 2). The uncertainty was acceptable. ENS and R2 were 0.71 and 0.75, and 0.71 and 0.76 during calibration and validation periods, respectively, demonstrating that the GLUE method can be used for runoff simulation and uncertainty analysis of the Xiaoqing River.

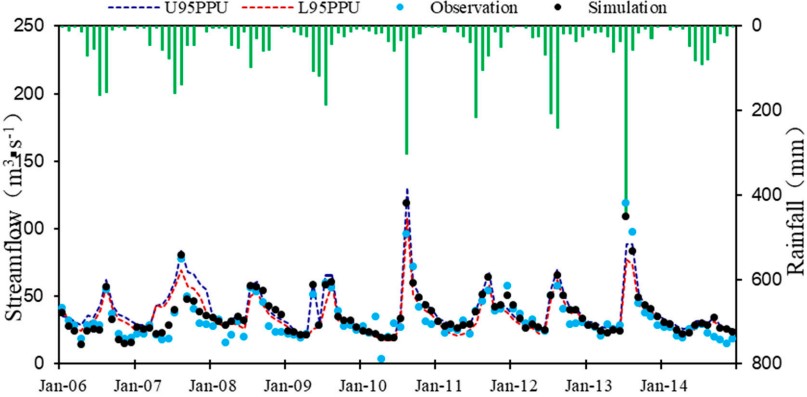

**Figure 4.** Runoff simulation of the Xiaoqing River basin in calibration and validation periods 95PPU based on SUFI-2.

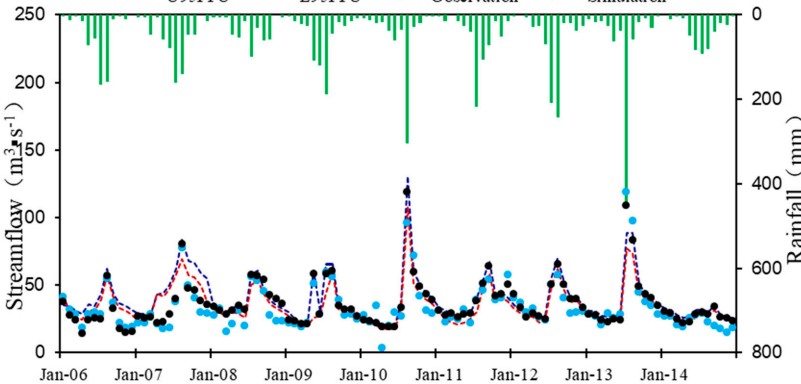

**Figure 5.** Runoff simulation of Xiaoqing River basin in calibration and validation periods 95PPU based on GLUE.

## 4.2. *Uncertainty Analysis of the Xinxue River*

### 4.2.1. Parameter Selection and Scale Determination

Six parameters were chosen, based on the analysis of sensitivity of the model and extensive practical research, together with the real circumstances of the Xinxue River (Table 3). SUFI-2 was iterated three times and 500 simulations were performed for each iteration. In GLUE, the objective function $E_{NS}$ was set as 0.6 with 3000 simulations. Further research and parameter uncertainty analysis were based on the last value range.

**Table 3.** Sensitivity parameter and value range of the Xinxue River drainage basin.

| Parameter | Parameter Meaning | Range of Initial Value | Influenced Object & Process |
|---|---|---|---|
| CN2 | SCS Runoff Coefficient | 35–98 | Surface Runoff |
| ALPHA_BF | Base Flow Coefficient $\alpha$ | 0–1 | Subterranean Water |
| GW_DELAY | Subterranean Water Lag Coefficient | 0–500 | Process of Subterranean Water |
| ESCO | Soil Evaporation Compensation Coefficient | 0.01–1 | Soil Evaporation |
| SOL_AWC | Available Water in Soil | 0–1 | Soil Moisture |
| CH_N2 | Drainage Line Manning Coefficient | −0.01–0.3 | Concentration of Channel |

### 4.2.2. Uncertainty Analysis of Model Parameter

When SUFI-2 was applied to the uncertainty analysis of the Xinxue River drainage basin, the scattered distribution patterns of $E_{NS}$ of CN2, ESCO, and CH_N2 became obvious. When CN2 increased, $E_{NS}$ first grew and then reduced (Figure 6). As ESCO rose, the uncertainty increased. When CH_N2 was added, the uncertainty reduced. The results indicated that the parameters CN2, ESCO, and CH_N had a great impact on the uncertainty of model simulation results, while the parameters ALPHA_BF, GW_DELAY, and SOL_AWC (which were well distributed) had little influence on it.

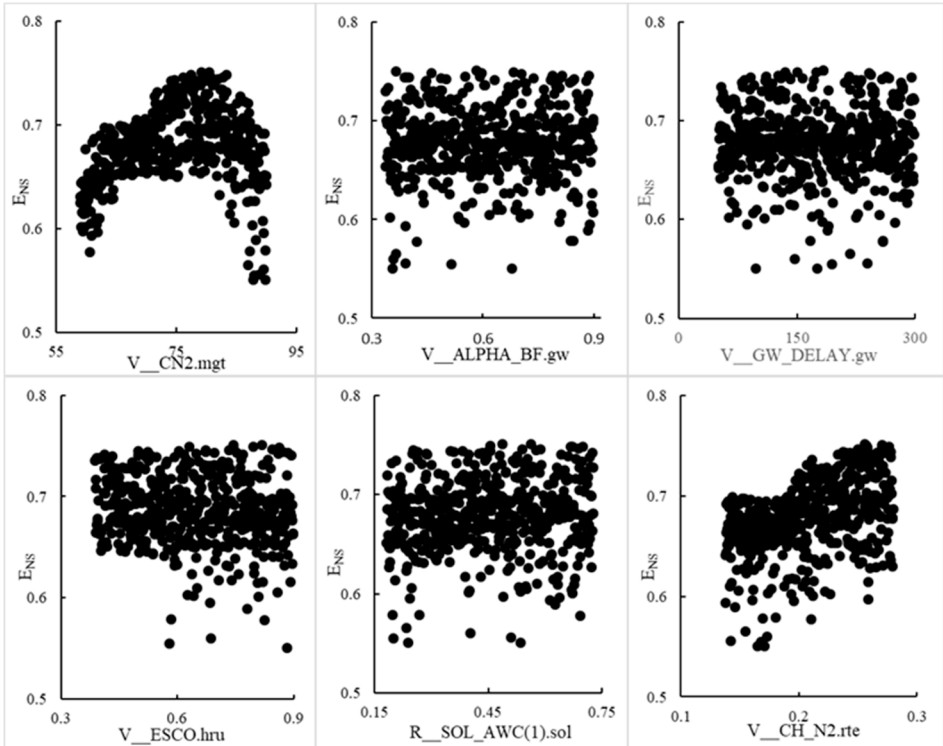

**Figure 6.** Parameter combination and distribution of Nash–Sutcliffe efficiency coefficient of Xinxue River basin based on SUFI-2.

When GLUE was applied to the uncertainty analysis of the Xinxue River drainage basin, the scattered $E_{NS}$ distribution patterns of V_CN2 and V_ALPHA_BF became obvious (Figure 7). As V_ALPHA_BF increased, the entirety first grew and then reduced. As V_CN2 increased, the entirety grew constantly and the uncertainty reduced. At the same time, uncertainty over the value of parameters V_GW_DELAY, V_ECSO, R_SOL_AWC, and V_CH_N2 was well distributed and there was no obvious change. This indicated that these four parameters had little influence on the results of model simulation.

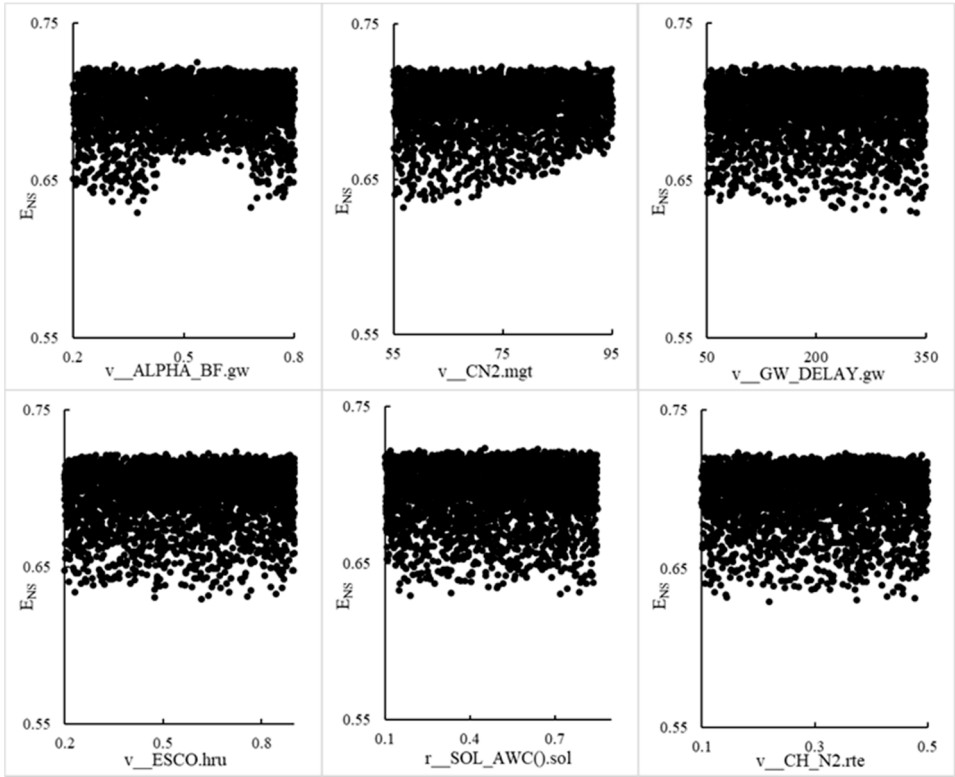

**Figure 7.** Parameter combination and distribution of Nash–Sutcliffe efficiency coefficient of Xinxue River basin based on GLUE.

### 4.2.3. Uncertainty Analysis of Model Simulation

The simulation results of the P-factor were 0.74 and 0.73 in calibration and validation periods, respectively, based on SUFI-2 (Figure 8, Table 4). Most measured values of the drainage basin were in the range of 95PPU. The R-factor was 0.87 and 0.75 in the calibration and validation periods, respectively, that is, values less than 1. $E_{NS}$ and $R^2$ were 0.71 and 0.73 in the calibration period, respectively, and 0.73 and 0.75 in the validation period, respectively. The results showed that SUFI-2 had a great effect on the verification and uncertainty analysis in the Xinxue River basin.

**Table 4.** Comparison of simulation results from the two methods.

| Method | Simulation | P-factor | R-factor | $E_{NS}$ | $R^2$ |
|---|---|---|---|---|---|
| SUFI-2 | calibration period | 0.74 | 0.87 | 0.71 | 0.73 |
| | validation period | 0.73 | 0.75 | 0.73 | 0.75 |
| GLUE | calibration period | 0.76 | 0.90 | 0.72 | 0.72 |
| | validation period | 0.73 | 0.83 | 0.73 | 0.74 |

The simulation results of the P-factor were 0.76 and 0.73 in the calibration and validation periods, respectively, based on GLUE (Figure 9, Table 4). Most measured values in the drainage basin were in the range of 95PPU. The R-factor was 0.90 and 0.83 in the calibration and validation periods, respectively, that is, values less than 1. The results show that SUFI-2 had a great effect on verification and uncertainty analysis in the Xinxue River basin. $E_{NS}$ and $R^2$ were 0.72 and 0.72 in the calibration period, respectively, and 0.73 and 0.74 in the validation period, respectively, which was satisfactory.

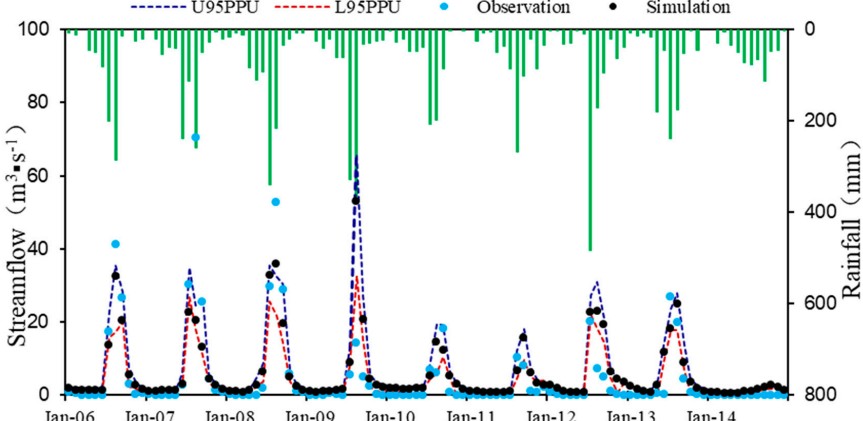

**Figure 8.** Runoff simulation of the Xinxue River basin in calibration and validation periods 95PPU based on SUFI-2.

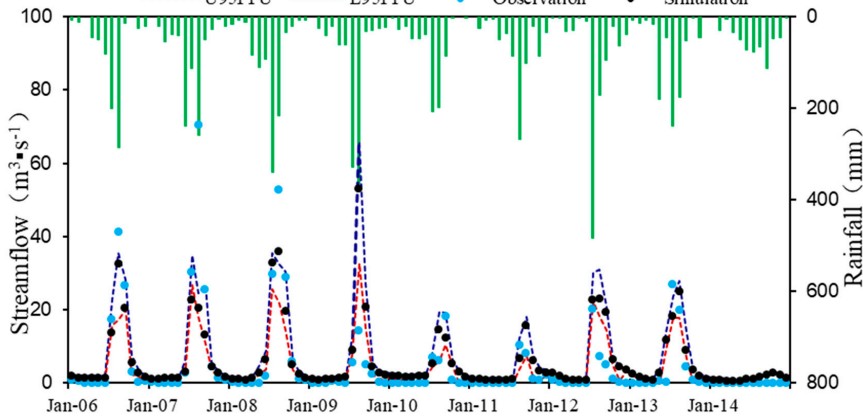

**Figure 9.** Runoff simulation of the Xinxue River basin in calibration and validation periods 95PPU based on GLUE.

## 5. Discussion

In simulations of the Xiaoqing River Basin, parameters CN2, ALPHA_BF, RCHRG_DP, V_GWQMN, and V_SOL_AWC had little influence on results. The parameter CN2 represents the runoff curves number of soil retention service (SCS) and defines the potential capacity of maximum soil moisture retention. The parameter ALPHA_BF refers to the base flow alpha factor, which defines the splitting factor of the base stream. The parameters CHRG_DP and V_GWQM represent the deep aquifer percolation fraction and threshold depth of water in the shallow aquifer required for return flow to occur, respectively. The parameter SOL_AWC is the effective water content of the soil. Narsimlu [26] used the SWAT model to perform simulation and uncertainty analysis on the Upper Sind River basin in the central plains of India. That study found that the parameters of CN2, SOL_AWC, ALPHA_BF, GWQMN, GW_REVAP, and others are highly uncertain; this also applies to the Xiaoqing River basin where the basin topography is dominated by plains. Therefore, we believe that the uncertainty of SWAT model parameters is affected by the geographical location of the basin. In addition, $E_{NS}$ varied

by parameter; this might have been because of the constant iterative optimization of parameter range by SUFI-2. That is why the GLUE method is better in reflecting the uncertainty of a single parameter. Most measured values were included in the 95PPU interval in the calibration and validation periods. In this sense, the GLUE method was better than the SUFI-2 method. With respect to the simulation effect, the $R^2$ of GLUE was 0.75 and 0.76 in the calibration and validation periods, respectively, values which were more than those of SUFI-2. Therefore, GLUE was better than SUFI-2 overall. In terms of calculation efficiency, the best simulation appeared when simulating for the 178th time using SUFI-2, while the best result appeared after simulating for 2364 times using GLUE. In this sense, the calculation efficiency of SUFI-2 was higher than that of GLUE. This was because LH random sampling was applied with SUFI-2 and there were fewer calculations. With respect to GLUE, however, Monte Carlo random sampling was applied and so there must be enough samples to get similar results. In this sense, GLUE needed longer to make its calculations.

In the simulation of the Xinxue River basin, parameters CN2, ESCO, CH_N2, and ALPHA_BF had a great influence on the model. The parameters ESCO and CH_N2 represent the compensation coefficient of soil evaporation and Manning's "n" value for the main channel, respectively. Narsimlu [57] uses the SWAT model and SUFI-2 method to perform simulation and uncertainty analysis on the Kunwari River basin, a small agricultural watershed in the hilly region of central India. The results show that the most uncertain parameters are CN2, ALPHA_BNK, ESCO, and CH_N2, and this agrees with the conclusion of our study. What differed was that the uncertainty of ESCO was greater, while the uncertainty of ALPHA_BF was less with SUFI-2 than with GLUE. Of the measured values in calibration and validation periods by SUFI-2 and GLUE, respectively, 70% were in the 95PPU interval. When comparing P-factor and R-factor in differing periods with SUFI-2 and GLUE, the uncertainty analysis of GLUE was slightly better than that of SUFI-2. With respect to the simulation effect, the $R^2$ of GLUE was 0.72 and 0.74 in the calibration and validation periods, respectively, less than those of SUFI-2 (0.73 and 0.75, respectively) (Table 4). Hence, the effects of SUFI-2 were better than those of GLUE. In terms of calculation efficiency, the best simulation was obtained when simulating for the 347th time. The best result was obtained after simulating 1789 times using GLUE, which demonstrates that the calculation efficiency of SUFI-2 was far higher than that of GLUE.

This research applied SUFI-2 and GLUE to do separate uncertainty analyses of the mesoscale Xiaoqing River basin in the plain, and the small-scale Xinxue River basin in the mountain area, and compared the two methods with regard to uncertainty of parameter, model prediction, and simulation effect. The results demonstrate that, in the parameters uncertainty analysis of the Xiaoqing River basin, the parameters CN2, ALPHA_BF, and RCHRG_DP had greater uncertainty with SUFI-2. CN2, GWQMN, and SOL_AWC had greater uncertainty with GLUE, which basically matches the conclusions of Narsimlu on the plain area. This is because the CN2, GWQMN, and SOL_AWC parameters are closely related to the properties of the underlying soil [58–60]. However, studies by Wilcke [61] and Ließ [62] show that the property of soil is greatly influenced by changes in terrain. We believe, therefore, that the parameter values of CN2, GWQMN, and SOL_AWC are closely related to the change in terrain. The Xiaoqing River basin has the typical topographical features of a plain basin, and so we felt it was more appropriate to apply GLUE in the Xiaoqing River basin. In terms of uncertainty of model prediction, GLUE can contain more measured values (large P value) as the confidence interval of uncertainty of the model is close (close in R value); this allows GLUE to have a better effect. In terms of simulation effect, the ENS values of SUFI-2 and GLUE are almost the same, although GLUE is better than SUFI-2. In this sense, we considered that GLUE is more satisfactory with regard to model simulation effect. We conclude that GLUE is more applicable in the Xiaoqing River basin; however, CN2, ESCO, and CH_N2 had greater uncertainty according to SUFI-2 while CN2 and ALPHA_BF had greater uncertainty according to GLUE. Our SUFI-2 results are close to those of Briak [63], who was studying a mountainous and hilly area. The Xinxue River basin is a small watershed in a mountain region with a relatively small drainage area, large river channel, high runoff convergence speed, and sudden changes in amounts of water. The parameters of CH_N are affected by watershed features and

have greater uncertainty, but the parameters of CN2 and ESCO have greater uncertainty because the terrain features of the Xinxue River basin had more influence. We consider that SUFI-2 analyzed the uncertainty of parameters more precisely. With respect to model prediction (R value and P value) and simulation effect ($E_{NS}$ and $R^2$), the results from SUFI-2 are all better than those of GLUE. So, overall, SUFI-2 is better than GLUE in the Xinxue River basin.

The results from the two methods in different basins lead us to consider that the analytical methods of uncertainty are influenced by basin features. This is because, as a method to further analyze the value range and distribution of model parameters, the analytical method of uncertainty is mainly influenced by parameters. Our analysis has proven that the uncertainty of parameters in the two basins is closely related to topography. In this sense, we believe that the analytical methods are connected to the terrain features.

## 6. Conclusions

In this study, two uncertainty analysis methods, SUFI-2 and GLUE, were studied by a distributed hydrological model to test their performance in model parameter uncertainty analysis. We studied their applicability in two watersheds with different characteristics. The SWAT model was used to conduct case studies in the Xiaoqing River basin and the Xinxue River basin in Shandong Province, China. The research results showed that the $E_{NS}$ of the monthly runoff simulation rate in the Xiaoqing River basin model was 0.71 and 0.74 in the periodic and verification periods, respectively, and $R^2$ was 0.72 and 0.76, respectively. In the Xinxue River basin model, $E_{NS}$ and $R^2$ were 0.71 and 0.73 in the periodic and verification periods, respectively, and the verification period was 0.73 and 0.75, respectively. The model had good applicability in both research areas.

In addition, GLUE and SUFI-2 were used to comprehensively analyze the parameter uncertainty, model prediction uncertainty, and model simulation effect of the SWAT model. The results show that GLUE was more suitable for the Xiaoqing River basin in the plain watershed. Compared with SUFI-2, GLUE had better simulation accuracy ($E_{NS}$ and $R^2$), and P-factor (the best coverage of measurement) value performed better when the R-factor (reasonably small uncertainty impacts) values were similar. In contrast, SUFI-2 was more suitable for the Xinxue River basin in the mountains. Compared with GLUE, SUFI-2 had better simulation accuracy ($E_{NS}$ and $R^2$), and the R-factor value of SUFI-2 was smaller when P-factor values were similar. In summary, we believe that the performance of the uncertainty analysis method was affected to some extent by the watershed characteristics. In the plain watershed, the GLUE method is more highly recommended, but in hilly areas, SUFI-2 is more suitable. In the future, we hope to carry out extended research, comparing and studying more uncertainty analysis methods in other watersheds with different characteristics, and therefore to contribute to research on the applicability of other uncertainty methods.

**Author Contributions:** L.Z. used a SWAT model for simulation, calibration, validation, and uncertainty analysis; Y.Y., G.X. and Z.L. collected and processed data; L.Z., B.X., and G.W. wrote the paper; W.S., H.S. and J.Y. supervised the research.

**Funding:** This research was funded by the National Natural Science Foundation of China (Nos. 31670451, 51679006, 51779007, 41671018), the Evaluation of Resources–Environmental Carrying Capacity in Typical Ecological Zones of Xinganling (Grant No. 12120115051201), and the Fundamental Research Funds for the Central Universities (No. 2017NT18).

**Conflicts of Interest:** The authors declare no conflict of interest.

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
