# Peer review of "Model Uncertainty Analysis Methods for Semi-Arid Watersheds with Different Characteristics: A Comparative SWAT Case Study"

_water, doi:10.3390/w11061177_

Round 1
Reviewer 1 Report
Dear Author
I must congratulate you on this manuscript. It is a very good attempt and you did a great job. Although there is not an innovative scientific contribution in terms of the methodology, the selected approach is in general well accepted in this community and the results can be useful in the region. There are certain shortcomings in this manuscript and I would like you to improve them at your earliest.
The first concern is the selection of these two methods of calibration, SUFI2, and GLUE. The selection has not been well justified and explained. The author did not provide enough information about how and why did they evaluate these two methods?
The second concern is I did not found any relative paragraph that explains the difference of calibration results while using these two methods.
The third concern is the discussion part that is quite short can be explained a bit more so that the reader can clearly understand.
The Conclusion part to me looks like a combination of results. There must be some sort of recommendation based on the authors findings. This section should be revised keeping in mind some solid recommendations.
Author Response
Responses to Reviewer 1:
We would like to thank you for the valuable questions which led to the improvement of the quality and presentation of this manuscript. We modified our manuscript on many of the points raised. Detailed replies to each comment are as follows:
1. The first concern is the selection of these two methods of calibration, SUFI2, and GLUE. The selection has not been well justified and explained. The author did not provide enough information about how and why did they evaluate these two methods?
Responses: Thanks a lot for your comments; We justified and explained these two methods by summarizing the uncertain analytical methods, introducing the SUFI-2 and GLUE methods in detail, and giving reasons for using these two methods in the introduction (Page 2, line 65-83).The SUFI-2 method is a comprehensive optimization and gradient search method, which not only can simultaneously determine multiple parameters, but also has the function of overall search. At the same time, it applies to models of complex structures considering the uncertainty of input data, model structures, parameters and measured data. The GLUE method combines the random sampling method of Monte Carlo and the Framework of Bayesian. This method is simple in principle, easy to operate and suitable for global analysis, which can also comprehensively analyze the model uncertainty caused by model structures, parameter values, data errors and other issues. What’s more, it is one of the widely used methods in the study of uncertainty in water environment models. Therefore, this study uses SUFI-2 and GLUE methods to research the uncertainty of the SWAT model.
2. The second concern is I did not found any relative paragraph that explains the difference of calibration results while using these two methods.
Responses: Thanks a lot for your comments; According to reviewer’s comments, we added a paragraph at the end of the discussion section to explain the difference of calibration results using these two methods (Page 14, line 410-443).
3. The third concern is the discussion part that is quite short can be explained a bit more so that the reader can clearly understand.
Responses: Thanks a lot for your comments; According to reviewer’s comments, we added the meaning of the parameters in the discussion section, and added references to verify that the uncertainty of the parameters is affected by the watershed characteristics. At the end of the discussion section, we explained the difference of calibration results using these two methods and gave reasonable advices based on the results of the discussion to perfect the whole discussion.
4. The Conclusion part to me looks like a combination of results. There must be some sort of recommendation based on the authors findings. This section should be revised keeping in mind some solid recommendations.
Responses: Thanks a lot for your comments. According to reviewer’s comments, we have re-edited the structures of the entire conclusion, and made reasonable suggestions based on the results of this paper. GLUE and SUFI-2 are used to comprehensively analyze the parametric uncertainty, uncertain model prediction and simulation effect of SWAT model. Our suggestion is GLUE is more suitable for the Xiaoqing River Basin in the plain watershed, but in the hilly area the SUFI-2 is better. According to the results of the two methods in differing basins, we further consider that the analytical methods of uncertainty are influenced by characteristics of watershed.

Reviewer 2 Report
Journal: Water
Manuscript ID: water-503326
Title: Model uncertainty analysis methods for semi-arid watersheds with different characteristics: a comparative SWAT case study
The authors have evaluated two of the widely used hydrological model uncertainty assessment tools at two heterogeneous catchments in China. The study is useful and is of interest for the global audience, however, in its current format, it is unsuitable for publication as it needs a major uplift in certain parts. Please see my comments below:
Keywords: the keywords should be different from the words used in the title
You introduced SWAT in line 40 without explaining the abbreviation and then again you have explained in line 44. Please correct this.
Ln 38-39: Please cite Deb et al. (2019) here as they also have shown model input data are limited by external factors.
Deb, P., Kiem, A.S. and Willgoose, G. Mechanisms influencing non-stationarity in rainfall-runoff relationships in southeast Australia. Journal of Hydrology. 2019, 571:749-764.
Ln 45: Expand RS and GIS as it's their first appearance.
Ln 50-51: I have no idea what are all these five acronyms used here. Please explain.
Ln 56-58: You claimed GLUE is more practical. What does it mean? Also, SUFI-2 is superior for models with complex structure. Give citations to these statements.
Why did you select only these two approaches (i.e. SUFI2 and GLUE) and not any other?
The introduction does not reflect the novelty of this study also it is too short. Please revise.
The climatic variables were collected for how many stations and where are they located? Mention this in the manuscript.
You did not mention anything about the model, model parameters and its structure. Discuss more!
How many iterations were done during model calibration?
Also, I believe section 3 (including 3.1 to 3.3) will be in the methods section as it describes the approach of model calibration and uncertainty analysis.
The discussion is weak. It needs to provide more details such as why certain parameters behaved in certain ways. Maybe because of the certain catchment characteristics? Also, the application of this study to global readers is unclear. You need to tailor your discussion in a way that it serves the purpose of showing the importance of this study to other regions as well.
Author Response
Responses to Reviewer 2: We would like to thank you for the valuable questions which led to the improvement of the quality and presentation of this manuscript. We modified our manuscript on many of the points raised. Detailed replies to each comment are as follows: 1. Keywords: the keywords should be different from the words used in the title Responses: Thank you for your advice. According to the comment, we re-edited the keywords section (Page 1, line 43-44). 2. You introduced SWAT in line 40 without explaining the abbreviation and then again you have explained in line 44. Please correct this. Responses: Thank you for your professional and careful correction. Due to the modification of the content of the article, we explained it in detail in the first place where SWAT appeared in this article, and deleted the repeated explanations in the text (Page 1, line 20). 3. Ln 38-39: Please cite Deb et al. (2019) here as they also have shown model input data are limited by external factors. Responses: Thank you for your professional advice. According to the comment, we have replaced the reference (Page 17, line 523-524). 4. Ln 45: Expand RS and GIS as it's their first appearance. Responses: Thank you for your professional and careful correction. We have modified it as a recommendation by the reviewer (Page 2, line 59). 5. Ln 50-51: I have no idea what are all these five acronyms used here. Please explain. Responses: Thank you for your careful review. According to the comment, We added the explain in detail of all these five acronyms. Since the previous content was modified, I have already explained SUFI-2, GLUE and MCMC when it appeared before. (Page 2, line 92-93). 6. Ln 56-58: You claimed GLUE is more practical. What does it mean? Also, SUFI-2 is superior for models with complex structure. Give citations to these statements. Responses: Thank you for your careful examination and patient guidance. GLUE is more practical mean for a simpler model, GLUE uncertainty analysis is more efficient compared with SUFI-2. This conclusion comes from Bhumika’s article “Parameter identification and uncertain analysis for simulating streamflow in a river basin of Eastern India” published in Hydrological Processes in 2015. The original conclusion is SUFI-2 is a semi-automatic optimized technique, which needs professional judgement. However, less number of simulation require to achieve a reasonable predicted uncertain ranges. Hence, it can greatly improve the operational efficiency of complex, structured and highly computational models. In contrast, GLUE is a fully automated and mighty optimized technique, which is easy to implement. Therefore, for a simpler model, GLUE is more practical. 7. Why did you select only these two approaches (i.e. SUFI2 and GLUE) and not any other? Responses: Thanks a lot for your comments. We justified and explained these two methods by summarizing the uncertain analytical methods, introducing the SUFI-2 and GLUE methods in detail, and giving reasons for using these two methods in the introduction (Page 2, line 65-83).The SUFI-2 method is a comprehensive optimization and gradient search method, which not only can simultaneously determine multiple parameters, but also has the function of overall search. At the same time, it applies to models of complex structures considering the uncertainty of input data, model structures, parameters and measured data. The GLUE method combines the random sampling method of Monte Carlo and the Framework of Bayesian. This method is simple in principle, easy to operate and suitable for global analysis, which can also comprehensively analyze the model uncertainty caused by model structures, parameter values, data errors and other issues. What’s more, it is one of the widely used methods in the study of uncertainty in water environment models. Therefore, this study uses SUFI-2 and GLUE methods to research the uncertainty of the SWAT model. 8. The introduction does not reflect the novelty of this study also it is too short. Please revise. Responses: Thank you for your advice. According to the comment, we have re-edited the introduction by adjusting the structures and summarize the uncertain analytical methods, and introducing the SUFI-2 and GLUE methods in detail. And we updated some of the literature, expanded the content of interview and pointed out the importance of this research at the end. Different from this innovative article, in the study of the existing model uncertainty, the contrast researches mainly focused on different kinds of uncertain analytical methods in the same basin, however, little research has concentrated on different watersheds compared different methods of uncertain analysis. Ignoring the impact of watershed characteristics on uncertainty analysis methods, therefore, our research content is that selecting two research areas with different watershed characteristics, and studying the applicability of uncertain analytical methods in different research areas. 9. The climatic variables were collected for how many stations and where are they located? Mention this in the manuscript Responses: Thank you for your careful review. We have re-edited the manuscript (Page 4, line 150-151). There are five meteorological stations (Jinan station、Huimin station、Kenli station、Weifang station、Yiyuan station) in Xiaoqing River basin and two meteorological stations (Xiji station、Shanting station) in Xinxue River basin. 10. You did not mention anything about the model, model parameters and its structure. Discuss more! Responses: Thank you for your careful examination and patient guidance. We added references to this section. The 3.1 chapter is an introduction and principle to the SWAT model structure in detail (Page 4, line 163-189). 11. How many iterations were done during model calibration? Responses: Thank you for your careful examination and patient guidance. We added this part to the manuscript, that SUFI-2 was iterated four times and 500 simulations were performed for each iteration. In GLUE, the objective function ENS was set as 0.6, and was iterated two times and with 3000 simulations were performed for each iteration (Page 6, line 250-251). 12. I believe section 3 (including 3.1 to 3.3) will be in the methods section as it describes the approach of model calibration and uncertainty analysis. Responses: Thank you for your professional advice. According to the comment, we re-edited the title in the manuscript (Page 4, line 162). 13. The discussion is weak. It needs to provide more details such as why certain parameters behaved in certain ways. Maybe because of the certain catchment characteristics? Also, the application of this study to global readers is unclear. You need to tailor your discussion in a way that it serves the purpose of showing the importance of this study to other regions as well. Responses: Thanks a lot for your comments. According to the comment, we added the meaning of the parameters in the discussion section, and added references to verify that the uncertainty of the parameters is affected by the watershed characteristics. At the end of the discussion section, we explained the difference of calibration results using these two methods and gave reasonable advices based on the results of the discussion. From this, we got a result that the performance of the uncertain analytical method is affected to some extent by the geographical location of the basin. In the plain watershed, the GLUE method is more recommended, but the SUFI-2 is better in the hilly area. According to the results of the two methods in differing basins, we further consider that the analytical methods of uncertainty are influenced by characteristics of watershed. All of the things we have been discussed above enrich the whole discussion.
Reviewer 3 Report
In this document, the comparison of the evaluation of the SWAT model in two watersheds was exposed. For that purpose, the authors compared the outcomes obtained by using GLUE and SUFI-2 methodologies for the performance of model parameters and uncertainty analysis. Even though the aim of the paper is scientifically interesting, the results and conclusions do not give any recommendation or suggestion for the replicability or relevance of this study in other watersheds. Consequently, from my humble point of view, this document needs a major revision, and special emphasis should be given to the relevance of the uncertainty analysis. Next, I give some points that I think can help to improve the document:
1.- Abstract: The structure should be revised. The categorical phrase of "the first to use two watersheds scale" is not true: there are dozens and hundreds of papers about SWAT model and its uses and applications.
2.- Introduction: This section is poor about examples. The authors should go deeper in the state -of-art in this field ...
3.- Materials: One watershed is located in a semi-arid and a semi-humid region at the same time, it sounds weird.
There are not any references for the model evaluation criteria: there area dozens of works about this topic, e.g., the famous Moriasi et al (2007) but there are lots of cases even more recent.
4.- Results: It is really hard to understand what is happening in figures 2 and 3. It seems that those results are not profusaly analyzed.
5.- Conclusions: are very general. It should include any applicability of this work in other watersheds, and future works (if possible).
Author Response
Responses to Reviewer 3: We would like to thank you for the valuable questions which led to the improvement of the quality and presentation of this manuscript. We modified our manuscript on many of the points raised. Detailed replies to each comment are as follows: 1. Abstract: The structure should be revised. The categorical phrase of "the first to use two watersheds scale" is not true: there are dozens and hundreds of papers about SWAT model and its uses and applications. Responses: Thank you for your professional and careful correction. According to reviewer’s comment, we modified the inappropriate statement and re-edited the entire abstract. 2. Introduction: This section is poor about examples. The authors should go deeper in the state -of-art in this field. Responses: Thank you for your advice. According to your comment, we have re-edited the introduction by adjusting the structures and summarize the uncertain analytical methods, and introducing the SUFI-2 and GLUE methods in detail. And we updated some of the literature, expanded the content of interview and pointed out the importance of this research at the end. The innovation of this article is in the study of the existing model uncertainty, the contrast researches mainly focused on different kinds of uncertain analytical methods in the same basin, however, little research has concentrated on different basins using different methods of uncertain analysis. Ignoring the impact of watershed characteristics on uncertainty analysis methods, therefore, our research content is that selecting two research areas with different watershed characteristics, and studying the applicability of uncertain analytical methods in different research areas. 3. Materials: One watershed is located in a semi-arid and a semi-humid region at the same time, it sounds weird. Responses: Thank you for your careful examination and patient guidance. This is our negligence in writing, we re-edited it in the manuscript (Page 3, line 138). 4. There are not any references for the model evaluation criteria: there area dozens of works about this topic, e.g., the famous Moriasi et al (2007) but there are lots of cases even more recent. Responses: Thank you for your careful examination and patient guidance. We added references to this section. The 3.1 chapter is an introduction to the SWAT model structure and principle in detail. At the same time, a lot of typical references also have been added, like Moriasi et al (2007) (Page 4, line 163-189). 5. Results: It is really hard to understand what is happening in figures 2 and 3. It seems that those results are not profusaly analyzed. Responses: Thanks a lot for your comments. Figures 2 and 3 represent the relationship between the Nash efficient coefficient and the sensitive parameter of the SUFI-2 method and the GLUE method, respectively. ENS is an indicator used to evaluate the results of hydrological model simulations. The bigger the ENS value is, the higher the matching degree between measured and simulated values is. According to reviewer’s comments, we have a more detailed analyses of this part (Page 7). 6. Conclusions: are very general. It should include any applicability of this work in other watersheds, and future works (if possible). Responses: Thanks a lot for your comments. According to reviewer’s comments, we have re-edited the structures of the entire conclusion, and made reasonable suggestions based on the results of this paper. GLUE and SUFI-2 are used to comprehensively analyze the parametric uncertainty, uncertain model prediction and simulation effect of SWAT model. Our suggestion is GLUE is more suitable for the Xiaoqing River Basin in the plain watershed, but in the hilly areas the SUFI-2 is better, we further consider that the analytical methods of uncertainty are influenced by characteristics of watershed. Moreover, we hope to carry out extended researches, like comparing and studying more kinds of uncertain analytical methods in various watersheds with different characteristics, and contributing the applicability of the researches to other uncertain methods.

Round 2
Reviewer 2 Report
Thanks for the revision. I feel the paper is now suitable for publication in Water in it's current format.
Reviewer 3 Report
The authors have considered and integrated the commentaries suggested.